# Temperature Effects during Impact Testing of a Two-Phase Metal-Ceramic Composite Material

**DOI:** 10.3390/ma12101629

**Published:** 2019-05-17

**Authors:** Eligiusz Postek, Tomasz Sadowski

**Affiliations:** 1Institute of Fundamental Technological Research, Polish Academy of Sciences, PL-02-106 Warsaw, Poland; 2Lublin University of Technology, PL-20-618 Lublin, Poland; t.sadowski@pollub.pl

**Keywords:** metal-ceramic composite, impact, coupled problem, thermomechanics

## Abstract

Metal-ceramic composite (MCC) materials can be used for manufacturing high-responsibility structures such as jet engines or cutting tools. One example of these materials is a two-phase wolfram carbide (WC) and cobalt (Co) composite. This MCC is a combination of hard WC grains with a Co metallic ductile binder. The resulting microstructure is a combination of two phases with significantly different mechanical behaviors. In this study, we investigate impact conditions, starting with an illustrative example of the Taylor impact bar where—although the process is very rapid—the equivalent plastic strain and temperature are higher in the adiabatic solution than those in the coupled solution. On exposing the WC/Co composite with a metallic binder to impact loading, heat is generated by plastic deformation. If the process is fast enough, the problem can be treated as adiabatic. However, a more common situation is that the process is slower, and the heat is generated in the ductile metallic binders. As a result, the associated grains are heated due to the conduction effect. Consequently, the process should be treated as coupled. When the impact is applied over a short time period, maximum temperatures are significantly lower if the process is analyzed as coupled rather than as adiabatic. The grains are immediately affected by temperature increase in the binders. Therefore, the heat conduction effect should not be omitted.

## 1. Introduction

Metal-ceramic composites (MCCs), such as wolfram carbide (WC)/cobalt (Co) composites, are used for manufacturing cutting tools. These composite materials are also applied in the aerospace and military industry. Structures made of MCCs are often exposed to impact loads and, hence, require computational models. 

The WC/Co composite material is made of high-strength WC grains embedded in a ductile Co matrix as a binder for joining the grains. Under impact load, plastic strains occur in the composite material. Since the process is rapid, the plastic strains occur in the binder. Some of the strains are converted into heat. The process has been thoroughly described for the Taylor bar experiment [1,2]. Two types of thermomechanical phenomena are distinguished: Adiabatic and coupled, as described in [3,4]. A preliminary investigation into coupling strategies is given in [5]. An extended numerical approach to thermomechanical coupling problems is proposed in [6,7]. A thermomechanical analysis, combined with investigation of second-order effects is given in [8,9]. Explicit solutions to the thermomechanical finite element discretized problems of metal forming and impact are proposed in [10,11]. A thermomechanical analysis of shell-layered structures is presented in [12,13].

Models of polycrystalline materials with elastic-viscous-plastic binders are analyzed in [14]. A detailed analysis of porous, ductile binders in polycrystalline materials is presented in [15]. The development of cracks around the junctions in polycrystalline materials is analyzed in [16]. The development of micro-cracking in polycrystalline material binders is investigated in [17]. Static and dynamic analyses of polycrystalline materials under pressure load are presented in [18]. The models of polycrystalline materials subjected to dynamic pressure and impact are qualitatively compared in [19].

The future aim of this study is to create a complex microstructural model made of two or more different phases that would describe the thermodynamical process of composite response in relation to time. Up to the authors’ knowledge, such a universal model has not yet been developed. This general model would allow for describing the following: Phase morphology; grain size distribution; interface length and thickness distribution; pore size distribution; pore placement inside the material; phase transformation; phase toughening; heat generation due to plastic work; micro-crack initiation process; micro-crack propagation mode (intergranular or transgranular); thermal stress evolution during cooling and solidification processes [20,21], as well as associated process of micro-cracks initiation and development.

In general, different types of multi-phase materials can be modeled, including basic internal microstructures consisting of two different types of grains: Brittle, semi-brittle, or elasto-plastic (e.g., Al_2_O_3_/ZrO_2_ [22,23,24,25,26,27], Al_2_O_3_/TiC, Al_2_O_3_/TiN [28,29]); elastic grains and elasto-plastic interfaces [14,15,16,17,18,19,30]; both elasto-plastic grains and interfaces fabricated from different materials.

The creation of a numerical model of a two-phase material requires the knowledge of several experimentally determined material parameters, estimated at the micro- and macro-level. These parameters describe both mechanical and thermal properties of the material and are investigated locally using real material samples. Moreover, macroscopic dynamic tests should be performed to verify the correctness of numerical assumptions.

The importance of composite materials under study results from their applications in different branches of engineering such as aerospace, mechanical, or civil engineering mainly due to safety reasons associated with the operation of critical structural parts under dynamic loading conditions. 

The aim of this paper is to create a numerical model of thermomechanical behavior of a polycrystalline material with ductile binders under impact loading conditions, with a focus on differences between adiabatic and coupled solutions. When composite materials with metallic binders are exposed to impact, heat is generated by plastic work. If the process is fast enough, the problem can be treated as adiabatic. However, a more common situation is that the process is slower with the heat generated in the ductile metallic binders, which means that the associated grains are heated due to the conduction effect. As a result, this process should be treated as coupled. When the impact is applied over a short time period, maximum temperatures are significantly lower when the process is analyzed as coupled rather than as adiabatic. The adjacent grains are immediately affected by temperature increase. A general conclusion resulting from the analysis is that the heat conduction effect should not be omitted, particularly during impact loading. We use the Abaqus code [31] to illustrate our considerations.

## 2. Numerical Model

### 2.1. Problem Statement

The thermal equilibrium equation is given in a finite element discretized form that satisfies the boundary and initial conditions. It reads as follows:(1)$$KT+C\dot{T}=F$$
where **T** and $\dot{T}$ denote the nodal temperature vector and the nodal temperature rate vector, respectively, **K** is the conductivity matrix, **C** is the heat capacity matrix, and **F** is the vector of thermal sources and fluxes. The mechanical problem is solved by the finite element method. It is given in a form suitable for explicit time integration as
(2)$$M\ddot{u}+D\dot{u}=f-p.$$

In the equation above (Equation (2)), **M** is the diagonal mass matrix, **D** is the damping matrix, **u** is the nodal displacement vector, $\dot{u}$ is the nodal velocity vector, $\ddot{u}$ is the nodal acceleration vector, **f** is the internal force vector, and **p** is the nodal loading vector. The formula below (Equation (3)) defines the **F** vector in Equation (1) due to plastic strain energy dissipation,
(3)$$f=\chi \sigma :e^{pl}$$
where *f* is the rate of heat generation, **σ** is the Cauchy stress tensor, **e***^pl^* is the rate of plastic deformation tensor, and *χ* is the Taylor–Quinney coefficient indicating the fraction of plastic work converted into heat [32]. In our considerations, we assume that it is only the heat flux that enters the **F** vector in Equation (1).

The Johnson-Cook plasticity model is given as follows [33],
(4)$$\sigma^{o}=\left(A+B{\left({\bar{\varepsilon }}^{pl}\right)}^{n}\right)\left(1+C\ln\left(\frac{{\dot{\bar{\varepsilon }}}^{pl}}{{\dot{\varepsilon }}_{0}}\right)\right)\;\left(1-\varsigma^{m}\right)$$
where *A* is the yield stress, *B* is the hardening coefficient, *C* is the strain rate coefficient, *n* is the hardening exponent, *m* is the thermal softening exponent, ${\bar{\varepsilon }}^{pl}$ is the equivalent plastic strain, ${\dot{\bar{\varepsilon }}}^{pl}$ is the rate of equivalent plastic strain, and ${\dot{\varepsilon }}_{o}$ is the reference strain rate. The dimensionless variable *ζ* describing the temperature effect is given in the following form,
(5)$$\varsigma =\left\{\begin{array}{ccc} 0 & for & T<T_{trans}\\ \left(T-T_{trans}\right)/\left(T_{melt}-T_{trans}\right) & for & T_{trans}\leq T\leq T_{melt}\\ 1 & for & T>T_{melt} \end{array}\right.$$
where *T* is the current temperature, *T_trans_* is the temperature around which the yield stress becomes temperature-dependent, and *T_melt_* is the melting temperature. 

### 2.2. Material Properties

The polycrystalline material under study consists of two phases: WC and Co. Given the significantly high strength of WC grains, the WC phase is ascribed the properties of a linear elastic material. The mechanical properties of this material are as follows: Young’s modulus—6.2 × 10^11^ Pa, Poisson’s ratio—0.215, thermal expansion coefficient—1.5 × 10^−5^ 1/K, mass density—14,770 kg/m^3^. Its thermal properties include a specific heat of 250 J/(kg·K) and a heat conductivity of 95 W/(m·K).

Experiments prove that the intergranular layers are elastic-plastic [34,35]. However, if we consider the impact conditions where some part of the plastic work is converted into heat, then a constitutive model of the material should be temperature-dependent. Co was described with the Johnson-Cook material properties [36]. In addition, it was assumed that the Taylor-Quinney coefficient *χ* is equal to 0.9 in both tested cases. Material properties of Co binders are given in Table 1.

The finite element analysis was performed using the Abaqus program [31], while the model was created with the MSC Patran program [37]. The postprocessing and visualization of results were performed with the use of the GiD program, [38]. 

## 3. Illustrative Example 

This example is derived from the Taylor bar test [39] analyzed by many researchers, for instance in [1,2]. We use this example to demonstrate the effects of heat conductivity and of solving the problem as coupled even though the impact process is very rapid. These assumptions are in contrast to the adiabatic solution. We compare adiabatic and coupled solutions of the same initial and boundary conditions. The bar material is assigned the same material properties as the binders in the polycrystalline material, analyzed in the next section.

A schematic design of the impact bar is shown in Figure 1. The bar hits a rigid obstacle with a velocity of 400 m/s. The fixed rigid plate is frictionless. The bar is discretized with 32,000 elements and 35,249 nodes. The adiabatic solution is modeled with C3D8R stress-displacement elements, while the coupled solution is modeled using C3D8RT elements with an additional degree of freedom for an unknown temperature at the nodes. Both element types are reduced integration elements. The length of the bar is 0.0324 m and the bar radius are set equal to 0.0032 m. Calculations are only made for half of the bar, considering the symmetrical boundary conditions. The time interval of the process is 0.8 × 10^−6^ s. 

Results are given in Figure 2. The adiabatic and coupled solutions of the problem are compared at the end of the analyzed time interval. It has been found that the solutions are different; even though the results are qualitatively similar, the maximum values of the field variables are different. The lowest difference between the adiabatic and coupled solutions can be observed for the maximum displacement, as shown in Figure 2a,b. The maximum displacement in the coupled solution is 0.4% lower than that obtained for the adiabatic solution. However, in the adiabatic solution, Figure 2c, the maximum equivalent plastic strain producing heat is 144% higher than that in the coupled solution, Figure 2e. This means that the maximum temperature of the bar is lower by 45.5% in the coupled solution. Based on the above example, we anticipate similar effects regarding the plastic strain and temperature of a polycrystalline material, which will be discussed further on. 

In the adiabatic case, the time step throughout the solution is varied from 8.411 × 10^−9^ s to 6.368 × 10^−9^ s. The critical time step ensuring stability of the explicit solution is evaluated at every time step. The end of the time interval is attained in 133,968 time steps. The solution takes 2120 s of a single core of the Intel^®^Xeon^®^ Processor E5 v3 (2,3 GHz 12-core (Haswell). In the coupled solution, the time step is varied from 8.411 × 10^−9^ s to 7.148 × 10^−9^ s. The end of the process is achieved in 131,391 steps, and the solution is obtained in 11,203 s of a single core of the same processor.

## 4. Analysis and Discussion of the Two-Phase Polycrystalline Material

The investigation concerned the WC/Co composite material shown in Figure 3. The polycrystalline material sample impacts a rigid wall with a velocity of 70 m/s, as shown in Figure 3a. The rigid wall is frictionless. The finite element discretization is shown in Figure 3b. The behavior of WC grains and Co binders is analyzed. We consider adiabatic and coupled solutions of the same initial and boundary conditions. The sample is discretized with 41,216 nodes. The grains and binders are discretized with 18,882 and 15,690 elements, respectively. In addition, Points A, B, C, and D are marked in Figure 3b. Variations in the equivalent plastic strain and temperature at the binder junctions marked with the letters are examined. The binders are assigned the same material properties as in the example discussed above.

The displacement fields and the deformed polycrystalline material are shown in Figure 4. The results demonstrate that the displacement fields in the adiabatic (Figure 4a) and the coupled solutions (Figure 4b) are qualitatively comparable. One can notice the formation of a wedge along the sliding lines created along the binders. The binders become thinner and are pressed out of the sample. The maximum displacement in the adiabatic case is 17% higher than that obtained in the coupled solution. It can also be noted that the deformed surface of the polycrystalline material in the coupled solution is smoother than that in the adiabatic solution, as shown in Figure 5.

The maximum Huber-Mises-Hencky (HMH) stress is slightly higher in the adiabatic solution than in the coupled one, as shown in Figure 6. The HMH stresses are 6.466 × 10^9^ Pa and 5.482 × 10^9^ Pa, respectively, with the difference amounting to 17.9%. It can be observed that the higher HMH stress regions are located towards the grain center. In terms of quality, the HMH stress distribution is more even in the coupled solution than in the adiabatic one, as shown in Figure 6a,b, respectively. 

Let us now examine the HMH stress distribution in the binders, as shown in Figure 7. The maximum HMH stresses are 2.245 × 10^9^ Pa and 4.593 × 10^9^ Pa in the adiabatic and coupled solution, respectively. The difference is considerable and amounts to 105%. An analysis of the data in Figure 8 reveals that the HMH stress is higher in the vicinity of the upper and lower edges of the binders. It can also be observed that the HMH stress is higher towards the grain interior. The cross-section is made along a segmented line behind the second row of the grains approx., counting from the hitting edge of the sample, where the impact has a significant effect on the inside section of the plate. For clarity, the binders were removed to make the grains visible. As a result, one can clearly see the higher HMH stress inside the grains.

Analyzing the HMH stress variations at junctions A, B, C, and D, it can be noted that at all examined junctions, the HMH stress is higher in the coupled solution than in the adiabatic one, as shown in Figure 9a,b, respectively. 

The distributions of the equivalent plastic strain are shown in Figure 10. The maximum equivalent plastic strain in the adiabatic solution is higher than that in the coupled solution, amounting to 3.204 and 1.742, respectively, as shown in Figure 10a,b. The difference is 83.9%. Also, the equivalent plastic strain is higher in the adiabatic solution at the marked junctions, as shown in Figure 11. Another interesting observation can be made with respect to the relationship between time and equivalent plastic strain at the analyzed junctions. The strain is very high at Junction C located at the tip of the created wedge, as shown in Figure 4. The equivalent plastic strain at Junction C is significantly higher than that at Junction B, although the latter is located closer to the hitting edge of the sample than the former. A more detailed distribution of the equivalent plastic strain in the binders is shown in Figure 12. The plots demonstrate that the equivalent plastic strain is higher in the vicinity of the upper and lower edges of the binder system. The same observation can be made with respect to the junctions in the binder system, as the equivalent plastic strains are concentrated there. It can be observed that Junction C is entirely plastic, and the equivalent plastic strain develops over the entire thickness of the plate.

Let us now pass to the temperature distributions shown in Figure 13. In the adiabatic solution, a temperature increase obviously occurs only in the binders, as shown in Figure 13a. The highest temperature is 1587 K, which means it is only 141 K below the melting point. In contrast to the adiabatic solution, the maximum temperature in the coupled solution is 382.6 K (Figure 13b), which is 315% lower than that achieved in the adiabatic solution. One can observe a qualitative difference between the two cases. Namely, an increase in temperature is also visible in the grains. For clarity, in Figure 14b and Figure 15a, the temperature distribution is shown separately for the binders and the grains. The maximum temperature in the binders is 381.55 K, which is only 12.8 K higher than the temperature value in the grains. 

3D temperature distributions in the binders are shown in Figure 14. In the adiabatic case, where the temperature increase only occurs in the binders, it can be observed that the temperature is higher in the vicinity of the binder edges and junctions, as shown in Figure 14a. This temperature distribution pattern corresponds to the regions of higher equivalent plastic strains. As for the coupled solution, the temperature distribution field is smooth; nevertheless, elevated temperatures can be observed in the corresponding regions of the binder system, as shown in Figure 14b.

Figure 15 shows the temperature increase in the grains observed for the coupled solution. The higher temperature in the grains is located near the binder junctions (Figure 15a) and in the vicinity of the grain edges, as shown in Figure 15b.

Another qualitative difference between the two cases can be spotted in Figure 16. A qualitative assessment of the relationship between time and temperature at the marked junctions demonstrates that the temperature at all junctions in the adiabatic case is higher than that measured in the coupled solution. The lowest temperature increase occurs at Junction D, where the equivalent plastic strain is the lowest of all four junctions, as shown in Figure 16b.

Another qualitative difference between the two tested solutions pertains to the time variation of temperature. At Junctions A and C, the temperature significantly increases at the beginning of the process but starts to decrease with time. This is the effect of heat conduction. In the adiabatic solution, the equivalent plastic strains at Junctions B and D are significantly lower than those at Junctions A and C, as shown in Figure 16a. With regards to the coupled solution, the temperature values at Junctions A, B, and C become similar towards the end of the observed time interval, as shown in Figure 16b. The temperature at Junction D remains close to 300 K in both solutions. This junction is located far from hitting edge of the sample and thus is not significantly affected by the impact.

In the adiabatic solution, the time step throughout the time domain varies from 6.998 × 10^−12^ s to 1.091 × 10^−11^ s. Similar to the illustrative example involving an impact bar, the critical time step is calculated at every time step. The end of the process is reached in 9182 time steps. The solution is reached in 171 s using the same processor as in the aforementioned example. In the case of the coupled solution, the solution time step remains in the same range as in the adiabatic case. The solution is obtained in 9182 time steps. The solution requires 206 s of a single core of the same processor. 

## 5. Conclusions

This paper examined the effects of heat generated by plastic work on a two-phase polycrystalline material consisting of elastic grains and elasto-plastic binders under impact conditions, focusing on the differences between the adiabatic and the coupled solution. Our findings are as follows: The displacement fields obtained for the two solutions are qualitatively similar; in terms of quality, the shape of the deformed sample surface is more uniform in the coupled solution than in the adiabatic case; although the HMH stress fields in both cases are identical with respect to quality, the stress field distribution in the coupled solution is more uniform. Moreover, the equivalent plastic strains in the binders are lower in the coupled solution; the maximum temperature is significantly lower in the coupled solution; the polycrystalline material grains are affected by temperature increase; given the significant differences in the solutions, the adiabatic solution should not be used for the analysis of polycrystalline materials.

The results of this work will be used in further research on a representative volume element of the polycrystalline material under impact conditions, including micro-crack initiation due to heat generation, with the application of a damage criterion (e.g., [17,30,40,41,42,43,44,45,46,47,48,49,50,51,52,53,54,55,56]).

## Figures and Tables

**Figure 1 materials-12-01629-f001:**
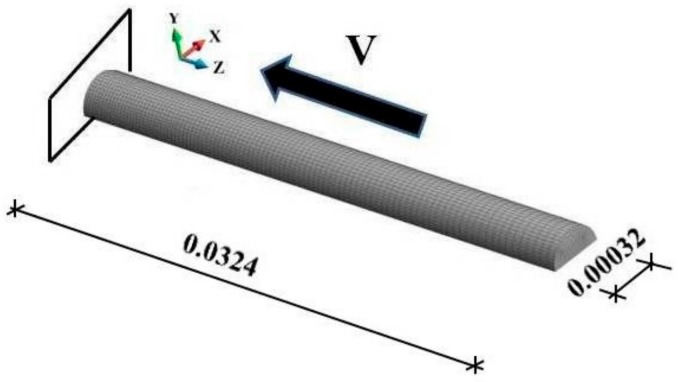
Impact scheme and finite element mesh of the bar (dimensions in m).

**Figure 2 materials-12-01629-f002:**
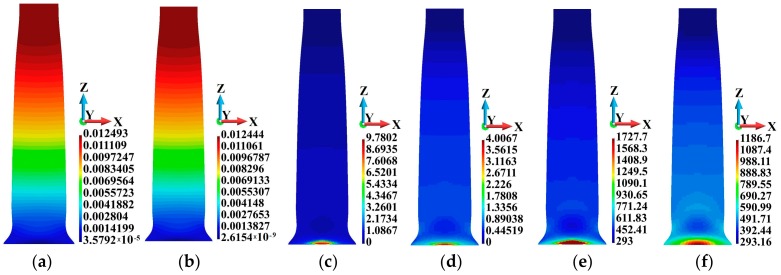
Comparison of the adiabatic and coupled solutions: (**a**) displacement field (m), adiabatic solution; (**b**) displacement field (m), coupled solution; (**c**) equivalent plastic strain field (non-dimensional), adiabatic solution; (**d**) equivalent plastic strain field (non-dimensional), coupled solution; (**e**) temperature field (K), adiabatic solution; (**f**) temperature field (K), coupled solution.

**Figure 3 materials-12-01629-f003:**
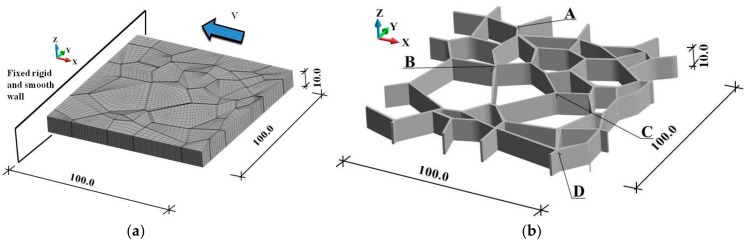
Scheme of the numerical model (dimensions in μm): (**a**) geometry, finite element discretization, boundary, and initial velocity conditions; (**b**) binder system and examined junctions.

**Figure 4 materials-12-01629-f004:**
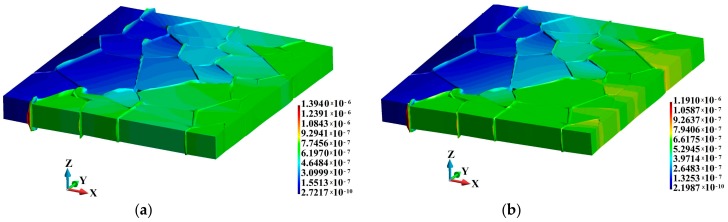
Displacement (m) fields (magnified 5 times): (**a**) Adiabatic solution; (**b**) coupled solution.

**Figure 5 materials-12-01629-f005:**
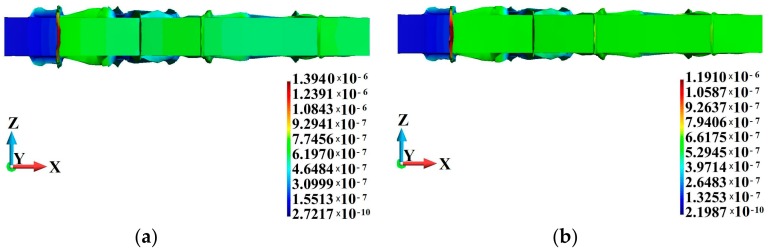
Displacement (m) fields (magnified 5 times), side view: (**a**) Adiabatic solution; (**b**) coupled solution.

**Figure 6 materials-12-01629-f006:**
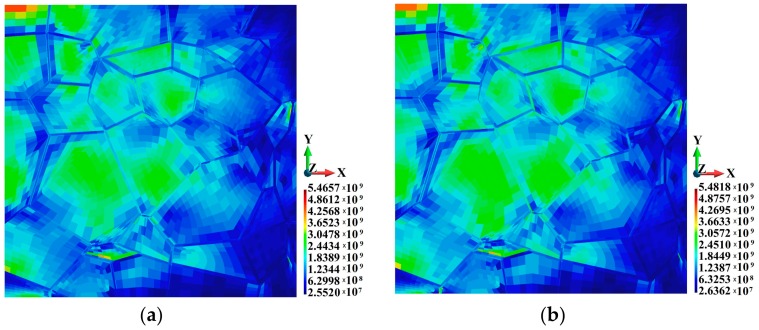
Huber-Mises-Hencky stress (Pa): (**a**) Adiabatic solution; (**b**) coupled solution.

**Figure 7 materials-12-01629-f007:**
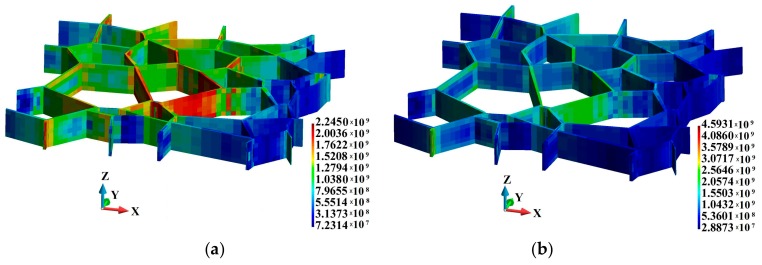
Huber-Mises-Hencky stress (Pa) in the binders: (**a**) Adiabatic solution; (**b**) coupled solution.

**Figure 8 materials-12-01629-f008:**
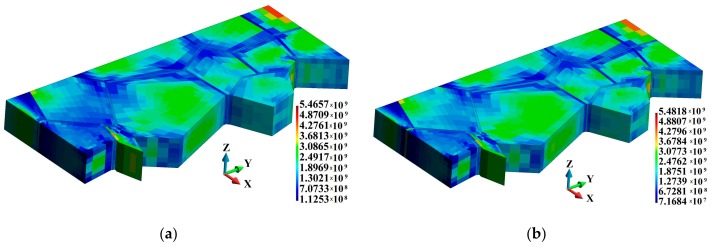
Huber-Mises-Hencky stress (Pa) in the cross-section along the grains: (**a**) Adiabatic solution; (**b**) coupled solution.

**Figure 9 materials-12-01629-f009:**
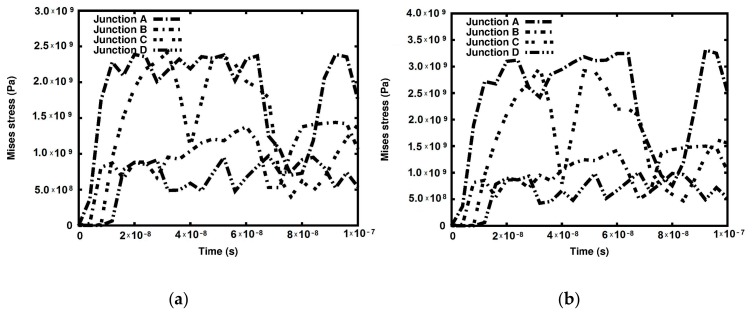
Variations in the Huber-Mises-Hencky stress (Pa) at the junctions: (**a**) Adiabatic solution; (**b**) coupled solution.

**Figure 10 materials-12-01629-f010:**
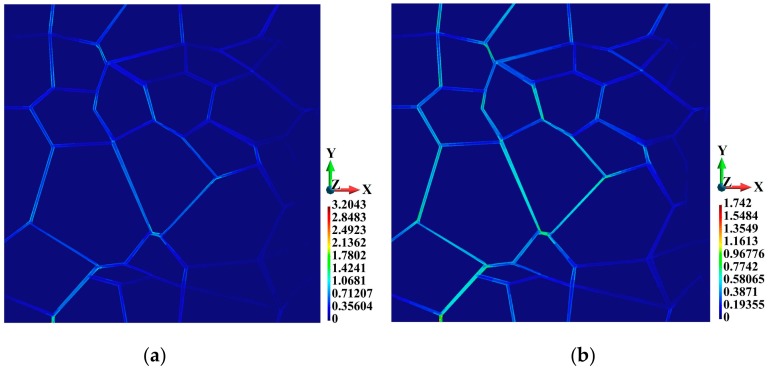
Equivalent plastic strain (non-dimensional), top view: (**a**) Adiabatic solution; (**b**) coupled solution.

**Figure 11 materials-12-01629-f011:**
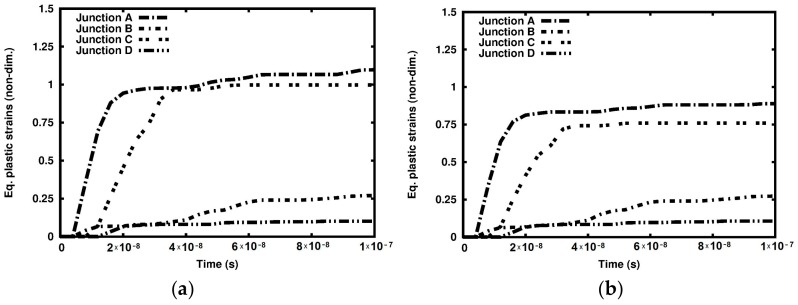
Equivalent plastic strain variations at the junctions: (**a**) Adiabatic solution; (**b**) coupled solution.

**Figure 12 materials-12-01629-f012:**
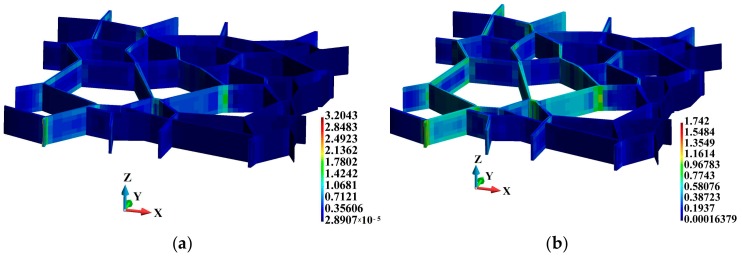
Equivalent plastic strain (non-dimensional) inside the binders: (**a**) Adiabatic solution; (**b**) coupled solution.

**Figure 13 materials-12-01629-f013:**
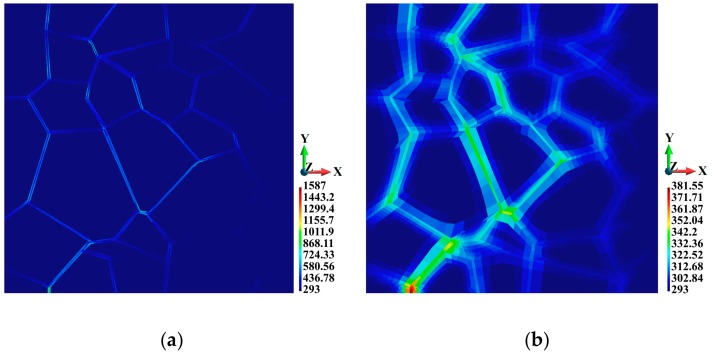
Temperature distribution (K), top view: (**a**) Adiabatic solution; (**b**) coupled solution.

**Figure 14 materials-12-01629-f014:**
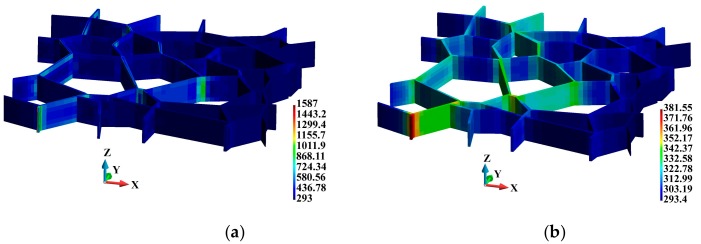
Temperature distribution (K) in the binders: (**a**) Adiabatic solution; (**b**) coupled solution.

**Figure 15 materials-12-01629-f015:**
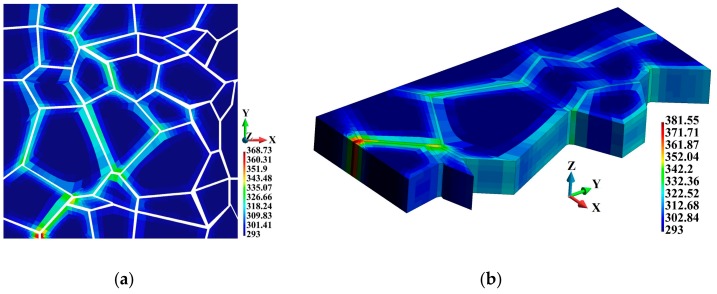
Temperature distribution (K), coupled solution: (**a**) Grains; (**b**) cross section along the grains.

**Figure 16 materials-12-01629-f016:**
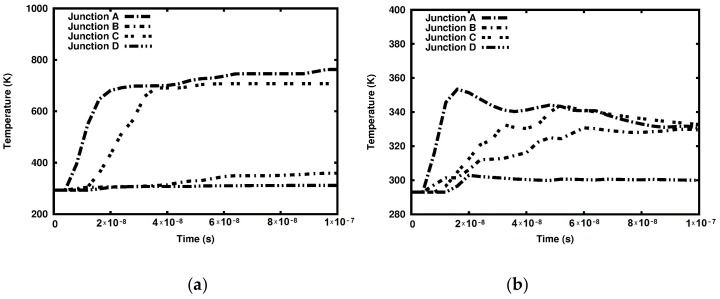
Temperature (K) variations at the junctions: (**a**) Adiabatic solution; (**b**) coupled solution.

**Table 1 materials-12-01629-t001:** Parameters of the Johnson-Cook material model for cobalt.

Parameters	Values and Units
Young’s modulus	2.1 × 10^5^ MPa
Poisson’s coefficient	0.296
Yield stress (A)	455.0 MPa
Hardening coefficient (B)	2475.0 MPa
Hardening exponent (n)	0.9
Strain rate coefficient C	0.0235
Thermal softening exponent (m)	1.0
Melt temperature T_melt_	1728 K
Transition temperature T_trans_	293 K
Specific heat C_H_	440 J/(kg·K)
Thermal conductivity K	150 W/(m·K)
Thermal expansion α_t_	5.0 × 10^−5^ 1/K
Mass density	9130 kg/m^3^

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
