# Peer review of "Temperature Effects during Impact Testing of a Two-Phase Metal-Ceramic Composite Material"

_materials, 2019, doi:10.3390/ma12101629_

Reviewer 1 Report

The introduction part is too general. The format is not correct.

The authors should provide boundary conditions and provide 3D results.

what unit for figure 2?

The author should reorganize the writing that can be readable for readers.

Author Response

Dear Reviewer,

We thank you for your important comments and we amended the manuscript. The answers for your specific remarks is given below.

With kind regards,

Authors

The introduction part is too general. The format is not correct.

Re: in the SI in Materials titled: *Behavior of Metallic and Composite Structures*  the presented manuscript plays the role of “Future paper”. Therefore, in the “Introduction”, we sketched a creation of the general micromechanical elasto-plastic or elastic 2 phase composite with substantially different properties.  The aim of the model is a description  of  thermodynamical processes in composites response related to time. In the several authors’ papers, different composites were described, partly due to quasi-static mechanical loading (ceramic matrix composites) or due to dynamic response of  WC/Co having hard elastic grains and plastic interfaces.

In the „Introduction”, we collected several microstructural data necessary to describe polycrystalline microstructure and other important processes which develops inside of material due to thermodynamical processes like: phases transformation and toughenning,   microcracks initiation and growth, as well as heat generation due to plastic work. In this way, we underlined general formulation of the future complex composite model.

In particular, in the present paper we limited to small part of the above general formulation, dealing with temperaturę field generation due to impact of metal-ceramic composite WC/Co.

In the next papers, we develop our conceptions for other types of multi-phase composites subjected to different types of mechanical or thermal loadins.  

The authors should provide boundary conditions and provide 3D results.

Re: we provided boundary conditions in the text below Figure 1 commenting that the fixed rigid plate is frictionless. We commented the boundary conditions in Figure 2 (a). We gave a comment in the text as well.

We provided the 3D results in the text and explained them. It gives further important observations and improves the manuscript.

What unit for figure 2?

Re: we added the units in Figure 2.

The author should reorganize the writing that can be readable for readers.

Re: we improved the writing including English language by native speaker.

Reviewer 2 Report

The paper proposes the coupled thermo-mechanical analysis, during impact, of a two-pahse wolfram carbide grains with cobalt metallic binders structure. Verification of the formulation and demonstration of its performance is done through a set of examples. The topic is of interest for the research community in this field of work. The paper is well organized and the presentation is of good quality, but some improvements are necessary.
- In the Introduction section it is written "The aim of the paper is creation [...]". It should be rewritten "The aim of the paper is the creation [...]" or "The aim of the paper is the developement [...]".
- In the Introduction section it is written "The aim of the paper is creation [...]", this paragraph should be moved at the end of the introduction section, in order to clarify to the readers what are the objectives of the present paper. Moreover, the authors should add in this paragraph if they used an in-house code or Abaqus, as they mentioned at the end of the section "2. Numerical model".
- In the introduction section, the reviewer suggests to enrich the bibligraphy with some recent developments dealing with the thermo-mechanical couplings:
Cinefra M, Valvano S, Carrera E (2016), ”Thermal stress analysis of laminated structures by a variable kinematic MITC9 shell element”, Journal of Thermal Stresses 39(2): 121-141
DOI: http://dx.doi.org/10.1080/01495739.2015.1123591
Carrera E, Valvano S (2017), ”A variable kinematic shell formulation applied to thermal stress of laminated structures”, Journal of Thermal Stresses 40(7): 803-827
DOI: http://dx.doi.org/10.1080/01495739.2016.1253439
- In the Introduction section it is written "Up to the authors knowledge such a universal model does not exists.". The definition "universal" concerning a theoretical model is too strong. There is always some aspect that it is simplified or not taken into account. The reviewer suggests to list all the future aspects that they wanted to study and if really these topics are not yet taken into account in a work, it should be underlined.
The reviewer suggest a general check of the manuscript. After this minor revision, the paper could be considered for publication.

Author Response

Dear Reviewer,

We thank you for your valuable remarks and we amended the manuscript. The answers for your specific remarks is given below.

With kind regards,

Authors

The paper proposes the coupled thermo-mechanical analysis, during impact, of a two-pahse wolfram carbide grains with cobalt metallic binders structure. Verification of the formulation and demonstration of its performance is done through a set of examples. The topic is of interest for the research community in this field of work. The paper is well organized and the presentation is of good quality, but some improvements are necessary.
- In the Introduction section it is written "The aim of the paper is creation [...]". It should be rewritten "The aim of the paper is the creation [...]" or "The aim of the paper is the developement [...]".

Re: we included proposed changes.
- In the Introduction section it is written "The aim of the paper is creation [...]", this paragraph should be moved at the end of the introduction section, in order to clarify to the readers what are the objectives of the present paper. Moreover, the authors should add in this paragraph if they used an in-house code or Abaqus, as they mentioned at the end of the section "2. Numerical model".

Re: we stated in the Introduction that we used Abaqus code. We rearranged the Introduction.
- In the introduction section, the reviewer suggests to enrich the bibligraphy with some recent developments dealing with the thermo-mechanical couplings:
Cinefra M, Valvano S, Carrera E (2016), ”Thermal stress analysis of laminated structures by a variable kinematic MITC9 shell element”, Journal of Thermal Stresses 39(2): 121-141
DOI: http://dx.doi.org/10.1080/01495739.2015.1123591
Carrera E, Valvano S (2017), ”A variable kinematic shell formulation applied to thermal stress of laminated structures”, Journal of Thermal Stresses 40(7): 803-827
DOI:
http://dx.doi.org/10.1080/01495739.2016.1253439

Re: we added the suggested valuable papers in the bibliography.
- In the Introduction section it is written "Up to the authors knowledge such a universal model does not exists.". The definition "universal" concerning a theoretical model is too strong. There is always some aspect that it is simplified or not taken into account. The reviewer suggests to list all the future aspects that they wanted to study and if really these topics are not yet taken into account in a work, it should be underlined.

Re: in the SI in Materials entitled: *Behavior of Metallic and Composite Structures*  the presented manuscript play the role of “Future paper”. Therefore in the “Introduction” we sketched creation of the general micromechanical elasto-plastic or elastic 2 phase composite with substantially different properties.  The aim of the model is a description  of  thermodynamical processes in composites response related to time. In the several authors’ papers, different composites were described, partly due to quasi-static mechanical loading (ceramic matrix composites) or due to dynamic response of  WC/Co having hard elastic grains and plastic interfaces.

In the „Introduction”, we collected several microstructural data necessary to describe polycrystalline microstructure and other important processes which develops inside of material due to thermodynamical processes like: phases transformation and toughenning,   microcracks initiation and growth, as well as heat generation due to plastic work. In this way, we underlined general formulation of the future complex composite model.

In particular, in the present paper, we limited to small part of the above general formulation, dealing with temperaturę field generation due to impact of metal-ceramic composite WC/Co.

In the next papers, we develop our conceptions for other types of multi-phase composites subjected to different types of mechanical or thermal loadins.  

The reviewer suggest a general check of the manuscript. After this minor revision, the paper could be considered for publication.

Re: The general checking was done and language was improved by native speaker

Round  2

Reviewer 1 Report

1. Please cite:

Characteristics of Thermal Stress Evolution During the Cooling Stage of Multicrystalline Silicon.

Lowering Dislocation Density of Directionally Grown Multicrystalline Silicon Ingots for Solar Cells by Simplifying Their Post-Solidification Processes—A Simulation Approach

2. the introduction and conclusion format is not correct. please use long sentence instead of dot sentence.

3 please provide all units in all figures.

4. the author should clarify the control group and different cases conditions.

Author Response

Dear Editor of Materials,

Please find enclosed improved version of my and Dr E.Postek article entitled:

“Temperature Effects During Impact Testing of a Two-Phase Metal-Ceramic Composite Material

I kindly ask you to consider our manuscript for possible publication in SI:

“Behaviour of of Metallic and Composite Structures” in the Materials, edited by me.

With kindest regards,

Tomasz Sadowski
